# Rapid respiratory cryptococcosis detection using targeted next-generation sequencing

Chaowen Deng[ID]1*, Miaoling Qiu2, Qingyan Yang3, Lina Li3, Baoling Liu1, Jieling Liu1, Ricky Wing-Tong Lau1, Fanfan Xing1

1 Department of Infectious Diseases and Microbiology, The University of Hong Kong–Shenzhen Hospital, Shenzhen, Guangdong, China, 2 Department of Hematology, The University of Hong Kong–Shenzhen Hospital, Shenzhen, Guangdong, China, 3 Department of Neurology, The University of Hong Kong–Shenzhen Hospital, Shenzhen, Guangdong, China

* dengchaowen08@163.com

## Abstract

*Cryptococcus neoformans*, a WHO critical-priority and neglected fungal pathogen, causes cryptococcosis primarily via inhalation. Clinical manifestations range from asymptomatic infection to fatal meningitis; suboptimal diagnostics cause delayed detection and high meningitis mortality. This retrospective study evaluated the clinical utility of targeted next-generation sequencing (tNGS) for diagnosing pulmonary cryptococcosis using respiratory specimens, compared to serum cryptococcal antigen (CrAg) testing and fungal culture. Over 38 months, 39 patients with confirmed cryptococcosis were enrolled. tNGS detected *Cryptococcus neoformans* in 92.3% (36/39) of respiratory samples, significantly outperforming fungal culture (23.1%, 9/39; $P < 0.05$). Serum CrAg testing was positive in 64.7% (22/34) of tested patients. tNGS identified *C. neoformans* in 12 CrAg negative cases and 75% (9/12) of culture-negative cases, enhancing diagnostic yield. The median turnaround time for tNGS (23 [IQR: 21–43] hours) was substantially shorter than for fungal culture (112 [IQR: 77.5–120.5] hours; Mann-Whitney U test, $P < 0.05$). Higher tNGS read counts correlated with culture positivity ($P < 0.05$) but not with CrAg status. Polymicrobial infections were detected in 76.9% (30/39) of tNGS tests, underscoring its utility in comprehensive pathogen identification. tNGS of respiratory specimens demonstrates superior sensitivity and markedly shorter turnaround time than fungal culture for cryptococcosis diagnosis. Crucially, its ability to detect *C. neoformans* in serum CrAg negative patients enables earlier diagnosis in cases missed by standard serology.

## Author summary

In this study, we assessed tNGS for diagnosing pulmonary cryptococcosis against conventional methods. Analyzing 39 patients, tNGS detected *C. neoformans* in 92.3% of respiratory samples, far exceeding fungal culture. tNGS also

**Data availability statement:** The authors confirm that all data underlying the findings are fully available without restriction. All relevant data are within the manuscript and supplementary file.

**Funding:** This study was partly supported by Sanming Project of Medicine in Shenzhen, China (SZSM201911014 to FX), and the University of Hong Kong-Shenzhen Hospital in Shenzhen, China (HKUSZH202504071 to CD). The funders had no role in study design, data collection and analysis, decision to publish, or preparation of the manuscript.

**Competing interests:** The authors have declared that no competing interests exist.

identified the pathogen in some serum CrAg negative cases and culture negative cases, closing critical diagnostic gaps. tNGS results were available much faster than fungal culture. While tNGS sequence reads aligned with culture positivity, they did not correlate with CrAg results, suggesting tNGS independently reflects respiratory fungal burden. We recommend integrating tNGS into diagnostic workflows for suspected cryptococcosis, especially when CrAg or culture results are negative or delayed. This approach promises earlier detection, timely therapy, and improved outcomes.

## Introduction

*Cryptococcus neoformans*, classified as a critical priority fungal pathogen on the WHO Fungal Priority Pathogens List, exists in the environment as basidiomycetous, encapsulated yeast that reproduces asexually via budding [1]. The respiratory tract serves as the primary portal of entry, with infection occurring when individuals inhale aerosolized basidiospores into the lungs, leading to cryptococcosis. The clinical presentation of cryptococcosis is diverse, ranging from asymptomatic infection to severe disseminated disease, with the central nervous system and lungs being the primary sites of infection. Notably, cryptococcal infections in immunocompetent individuals often remain asymptomatic, resulting in delayed diagnosis until advanced stages [2,3]. Current clinical diagnosis of cryptococcosis relies on laboratory methods including India ink staining and Cryptococcal PCR of cerebrospinal fluid (CSF), fungal culture of clinical specimens, and cryptococcal antigen (CrAg) detection in serum or CSF; however, each with notable limitations [4,5]. For instance, fungal cultures of sterile body fluids exhibit low sensitivity, and CrAg testing is frequently omitted during early diagnostic evaluations [6]. Consequently, diagnostic delays often lead to meningitis and increased mortality.

In recent years, next-generation sequencing (NGS) has emerged as a powerful technology for rapidly identifying a broad range of pathogens directly from clinical samples, significantly enhancing diagnostic accuracy for various infectious diseases. Our previous studies have demonstrated its utility in confirming fungal infections, including those caused by *Talaromyces marneffei* and *Pneumocystis jirovecii*, as well as infections involving fastidious or slow-growing pathogens such as *Nocardia* spp., *Mycoplasma pneumoniae*, and *Coxiella burnetii* [7–11]. Furthermore, NGS has increasingly facilitated the diagnosis of cryptococcosis, enabling earlier detection of cases that might otherwise be missed [12–14]. While NGS has been acknowledged as a valuable diagnostic tool for infections, its clinical validity and impact on cryptococcosis patient management when using respiratory specimens remains uncertain.

This single-center retrospective study evaluated the diagnostic performance of serum CrAg, targeted NGS (tNGS), and fungal culture in patients with cryptococcosis over a 38-month period. We compared the diagnostic performance of three methods: serum CrAg testing, tNGS and fungal culture of respiratory specimens. Additionally, we assessed the time required to identify *C. neoformans* in respiratory specimens

using tNGS versus conventional fungal cultures. Furthermore, we investigated whether higher tNGS sequencing reads correlated with positive culture results and assessed the interplay between serum CrAg status and diagnostic outcomes. We anticipate these findings will contribute to more rapid and accurate diagnosis, thereby optimizing the clinical management of cryptococcosis.

## Materials and methods

### Ethical statement

This study received approval from the Medical Research Ethics Committee of The University of Hong Kong - Shenzhen Hospital ([2025]143) on 22nd May, 2025. All tNGS was performed on respiratory samples obtained from patients or their guardians with written informed consent.

### Patients

This study was conducted over a 38-month period, from January 2022 to February 2025, at The University of Hong Kong–Shenzhen Hospital in Shenzhen, China. Study included patients with confirmed cryptococcosis identified through hospital electronic clinical medical record system. All patients diagnosis of cryptococcosis was established according to the 2019 consensus guidelines of the European Organization for Research and Treatment of Cancer (EORTC) and the Mycoses Study Group Education and Research Consortium (MSGERC) [15]. The inclusion criteria of patients were as follows: (i) patients aged over 18 years; (ii) patient's respiratory specimen tested by tNGS; (iii) all specimens were collected prior to the initiation of cryptococcosis treatment. The exclusion criteria were: (i) patients with incomplete clinical data; (ii) patients who were lost to follow-up. Clinical characteristics, radiological findings, laboratory data, and clinical outcomes of the patients were collected and analyzed. The enrollment and screening process of patients is depicted in Fig 1.

### Microbiological methods

Clinical specimens were collected and processed following standard protocols [16]. For fungal culture, lower respiratory specimens, such as sputum and bronchoalveolar lavage (BAL), were each inoculated onto two Sabouraud dextrose agar (SDA) plates and one triphenyltetrazolium chloride SDA plate. One of the SDA plates and the triphenyltetrazolium chloride SDA plate were incubated at 37°C, and the other SDA plate was incubated at 25°C. The agar plates were examined daily for the first five days, and then twice weekly afterwards. All suspected colonies were identified based on morphological characteristics, conventional biochemical methods and matrix-assisted laser desorption/ionization-time of flight mass spectrometry (MALDI-TOF MS) Microflex LT/SH (Bruker Daltonics, Bremen, Germany). India ink staining of CSF revealed *C. neoformans* yeasts with thick capsules, highlighted by characteristic clear halos. Cryptococcal antigen detection was performed using lateral flow assay (Norman, USA) in serum and CSF. 1,3-β-D-glucan detection was performed using Test Kit for the Detection of Fungus 1,3-β-D-Glucan (Photometric Assay) (A & C Biological Ltd, Zhanjiang, China).

### Target Next-Generation Sequencing (tNGS)

*Sample preparation and nucleic acid extraction.* Lower respiratory samples (≥600 μL) were collected in sterile tubes, stored at 4°C, and transported within 8 hours for multiplex PCR-based tNGS analysis. Samples with high viscosity were diluted 1:1 with 0.1% (w/v) dithiothreitol before nucleic acid extraction. For homogenization, 600 μL of the sample and 1.5 g of glass beads were agitated at 4700 rpm for 135 seconds using the FastPrep-24 5G Instrument (MP Biomedical, CA, USA). Nucleic acids were extracted from 250 μL of the homogenized supernatant using the MagPure Pathogen DNA/RNA Extraction Kit (Magen Biotechnology, Guangzhou, China). To ensure assay quality, Jurkat cells (c101-b; IGE Biotechnology, Guangzhou, China) served as a negative control, while Jurkat cells spiked with *Bacillus subtilis* (Guangdong Microbial Culture Collection Center, Guangzhou, China) were used as a positive control



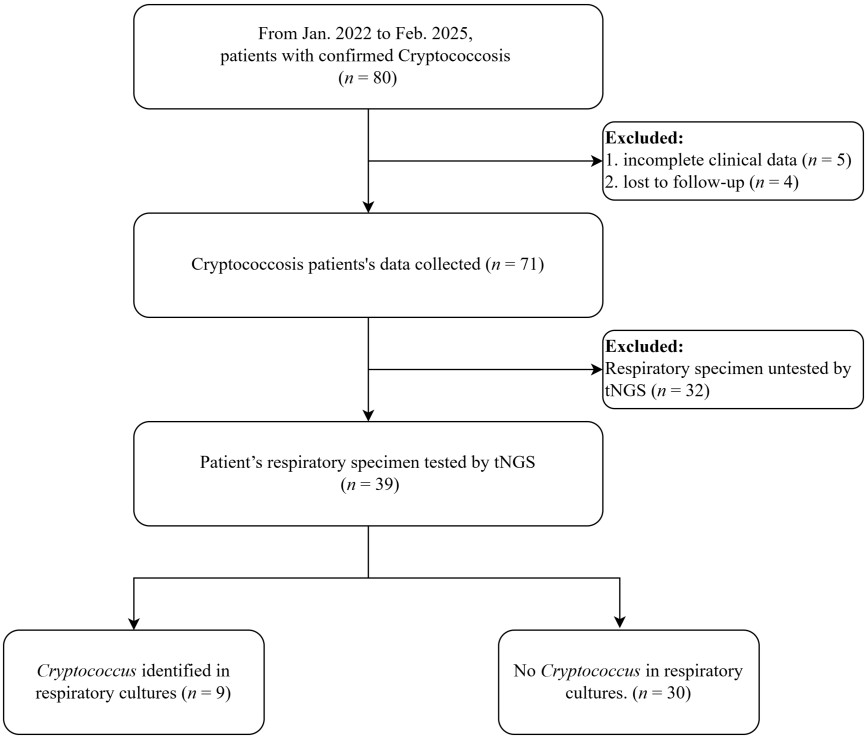

**Fig 1. Flowchart of cryptococcosis patient enrollment and grouping.** (tNGS: Targeted Next-Generation Sequencing).

**Library construction and sequencing.** cDNA synthesis, multiplex PCR preamplification of target loci, and library preparation (the minimum DNA input quantity of 1μg) were performed using the RP100 Respiratory Pathogen Multiplex Testing Kit (KingCreate, Guangzhou, China). This kit targets 198 pathogens for the early diagnosis of respiratory infections, including 80 bacteria, 79 viruses (35 DNA viruses and 44 RNA viruses), 32 fungi, and 7 mycoplasmas/chlamydia (a complete species are listed in S1 Data). Libraries were quantified using the Equalbit DNA HS Assay Kit (Vazyme, Nanjing, Jiangsu, China) with the Invitrogen Qubit 4.0 Fluorometer (Thermo Fisher, Waltham, MA, USA). Libraries with a typical fragment size of 250–350 bp and a minimum concentration of 0.5 ng/μL were pooled. The pool concentration was re-measured and diluted to a final working concentration of 1 nmol/L. For sequencing, 5 μL of the pooled library was denatured with 5 μL of 0.1 M NaOH, vortexed, centrifuged, and incubated at room temperature for 5 minutes. The diluted and denatured library was subsequently sequenced on an Illumina MiniSeq platform (CA, USA) utilizing a universal sequencing reagent kit (KS107-CXR, KingCreate, Guangzhou, China). On average, each library yielded approximately 0.1 million reads, with a sequencing read length of single-end 100 bp.

**Bioinformatics analysis and positivity thresholds.** Data analysis was conducted using the data management and analysis system (v3.7.2, KingCreate). Raw sequencing reads were processed with Fastp (v0.20.1) to trim adapters and filter low-quality data, retaining only reads with a length >50 bp and a Q30 score >75% [17]. High-quality reads were then aligned to a curated reference database (comprising GenBank, RefSeq, and NCBI Nucleotide entries) using Bowtie2 (v2.4.1) in "very sensitive" mode [18]. Taxonomic abundance was quantified as reads per 100,000 (RPhK) at the species and genus levels. Based on a previous retrospective study, positivity thresholds were defined as 7 RPhK for viruses, 15 RPhK for bacteria, and 11 RPhK for fungi [19]. The analytical LoD was determined by spiking microorganisms into Jurkat cells ($10^5$ cells/mL) in a serial 3-fold dilution series, ranging from 4,050 CFU/mL to 50 CFU/mL (or copies/mL for

viruses). For each concentration, four replicates were tested. The LoD was defined as the lowest concentration at which the pathogen was detected in all four replicates, meeting the following criteria in simulated specimens: an RPhK value ≥15 with amplicon coverage ≥1, or an RPhK value ≥45.

**Generation of tNGS results reports.** Microorganisms were classified as potential pathogens based on the following criteria: (i) bacteria (excluding the *Mycobacterium tuberculosis* complex): RPhK ≥ 15 with amplicon coverage ≥50%; (ii) viruses: RPhK ≥ 7 with amplicon coverage ≥50%; (iii) fungi: RPhK ≥ 11 with amplicon coverage ≥50%; and (iv) the *Mycobacterium tuberculosis* complex: RPhK ≥ 1 with amplicon coverage ≥50%. Yin et al. provided a complete and detailed description of the experimental principles, procedures and methodology of the tNGS tests used in this study [19]. The identified microorganisms and their corresponding sequence reads were reported to The University of Hong Kong - Shenzhen Hospital by the laboratory team. All tNGS was performed on respiratory samples obtained from patients or their guardians with written informed consent.

## Statistical analysis

Continuous variables, assessed for normal distribution, were compared using the unpaired t-test or the Mann-Whitney U test. A two-proportion Z-Test was used to compare the diagnostic sensitivity between the two detection methods. Statistical tests were performed using IBM SPSS software (version 26.0; SPSS Inc., Chicago IL, USA) and GraphPad software (version 9.3.1, GraphPad Software, San Diego, CA, USA) with a *P*-value < 0.05 as the significance threshold. This study is a non-interventional observational study. No sample size estimation, randomization, or blinding was performed.

## Results

### Patient characteristics

As displayed in Fig 1, 80 patients with confirmed cryptococcosis were enrolled between January 2022 and February 2025. After excluding 5 patients with incomplete clinical data, 4 lost to follow-up and 32 whose respiratory specimen lacked tNGS testing, 39 patients were included in the final analysis. Among those, participants with respiratory culture results were stratified by *C. neoformans* status: culture-positive (*n* = 9) and culture-negative (*n* = 30). *C. neoformans* was detected in 92.3% (36/39) of respiratory specimens, while no *C. neoformans* were identified in the remaining 7.7% (3/39).

This study comprised 66.7% (26/39) males, with a median age of 69 years (IQR: 42–83). Underlying comorbidities were present in 79.5% (31/39) of patients, the most frequent were diabetes (28.2%, 11/39), cancer (28.2%, 11/39), and autoimmune diseases (17.9%, 7/39). Despite this, 38.5% (15/39) were immunocompetent. The most common presenting symptom was cough (79.5%, 31/39), followed by sputum and fever. Diagnostic evaluation was primarily triggered by thoracic imaging: 56.4% (22/39) had pneumonia-like findings, and 41.0% (16/39) exhibited pulmonary nodules. Lumbar puncture to assess for cryptococcal meningitis was performed in only 48.7% (19/39) of patients; among these, no cases of CNS involvement were identified.

Antifungal therapy was administered to 94.9% (37/39) of patients. Monotherapy was utilized in 35 cases, while 2 received combination regimens. Clinical improvement was observed in 74.4% (29/39) of treated patients, with a mortality rate of 25.6% (10/39) (Table 1).

### Microbial identification using tNGS for respiratory specimens

*C. neoformans* was detected in respiratory specimens of 92.3% (36/39) patients by tNGS, suggesting high diagnostic sensitivity. Microbial findings from tNGS were categorized as follows: (i) exclusive *C. neoformans* detection (15.4%, 6/39); (ii) polymicrobial detection with viruses (56.4%, 22/39); (iii) polymicrobial detection without viruses (20.5%, 8/39); (iv) no *C. neoformans* identified (7.7%. 3/39) (Fig 2A). Among the 30 polymicrobial specimens (categories ii and iii), organisms frequently associated with respiratory infections or colonizing were identified, including *Acinetobacter baumannii* (20%, 6/30), *Klebsiella pneumoniae* (16.7%, 5/30), *Staphylococcus aureus* (33.3%, 10/30) and *Streptococcus anginosus* (26.7%,

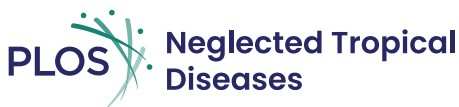

**Table 1. Clinical features of enrolled patients with cryptococcosis.**

| Characteristics | Number of patients n (% [95%CI]) | Patient's respiratory specimen tests by tNGS (*n*=36, 92.3% [95%CI]) | *Cryptococcus* was identified by fungal culture (*n*=9, 23.1% [95%CI]) |
|---|---|---|---|
| **Gender** | | | |
| Male | 26 (66.7 [50.9-79.4]) | 26 (72.2 [55.8-84.2]) | 7 (77.8 [44.1-94.3]) |
| Female | 13 (33.3 [20.6-49.1]) | 10 (27.8 [15.8-44.2]) | 2 (22.2 [5.7-55.9]) |
| **Age, years (IQR)** | 69 (42, 83) | 72 (41, 83) | 62 (40, 79) |
| **Underlying diseases** | | | |
| Diabetes | 11 (28.2 [16.5-43.9]) | 10 (27.8 [15.8-44.2]) | 3 (33.3 [12-64.9]) |
| Cancer | 11 (28.2 [16.5-43.9]) | 10 (27.8 [15.8-44.2]) | 3 (33.3 [12-64.9]) |
| Autoimmune diseases | 7 (17.9 [8.8-33.1]) | 7 (19.4 [9.6-35.4]) | 1 (11.1 [0.2-46]) |
| Hematologic malignancy | 1 (2.6 [0-14.6]) | 0 (0 [0-11.8]) | 0 (0 [0-35]) |
| HIV | 1 (2.6 [0-14.6]) | 1 (2.8 [0-15.7]) | 0 (0 [0-35]) |
| **Immunocompetent** | 15 (38.5 [24.9-54.2]) | 14 (38.9 [24.8-55.2]) | 3 (33.3 [12-64.9]) |
| **Symptoms** | | | |
| Fever | 21 (53.8 [38.6-68.4]) | 19 (52.8 [37-68]) | 4 (44.4 [19-73.3]) |
| Cough | 31 (79.5 [64.1-89.4]) | 28 (77.8 [61.6-88.4]) | 6 (66.7 [35.1-88]) |
| Sputum | 28 (71.8 [56.1-83.5]) | 25 (69.4 [53-82]) | 6 (66.7 [35.1-88]) |
| Headache | 2 (5.1 [0.6-18.0]) | 1 (2.8 [0-15.7]) | 1 (11.1 [0.2-46]) |
| **Indication for cryptococcosis work-up** | | | |
| Pulmonary nodules | 16 (41.0 [27.1-56.6]) | 16 (44.4 [29.6-60.4]) | 6 (66.7 [35.1-88]) |
| Pneumonia | 22 (56.4 [41-70.7]) | 20 (55.6 [39.6-70.4]) | 2 (22.2 [5.7-55.9]) |
| CNS infection | 1 (2.6 [0-14.6]) | 0 (0 [0-11.8]) | 1 (11.1 [0.2-46]) |
| **Lumbar puncture** | | | |
| Yes | 19 (48.7 [33.9-63.8]) | 17 (47.2 [32-63]) | 6 (66.7 [35.1-88]) |
| CNS involved | 0 (0 [0- 10.9]) | 0 (0 [0-11.8]) | 0 (0 [0-35]) |
| **Regimen** | | | |
| Monotherapy | 35 (89.8 [75.7-96.4]) | 32 (88.9 [73.9-96.1]) | 8 (88.9 [54-99.8]) |
| Combination therapy | 2 (5.1 [0.6-18]) | 2 (5.55 [0.7-19.3]) | 1 (11.1 [0.2-46]) |
| No treatment | 2 (5.1 [0.6-18]) | 2 (5.55 [0.7-19.3]) | 0 (0 [0-35]) |
| **Outcome** | | | |
| Improved | 29 (74.4 [58.7-85.5]) | 26 (72.2 [55.8-84.2]) | 8 (88.9 [54-99.8]) |
| Succumbed | 10 (25.6 [14.5-41.3]) | 10 (27.8 [15.8-44.2]) | 1 (11.1 [0.2-46]) |

Remarks: CNS, central nervous system; CSF, cerebrospinal fluid; IQR, inter quartile range.

8/30). Additional detected pneumonia-associated pathogens comprised *Escherichia coli*, *Haemophilus influenzae*, *Pseudomonas aeruginosa*, and *Streptococcus pneumoniae* (Fig 2B). These findings underscore tNGS's dual utility in diagnosing *C. neoformans* infections and identifying clinically significant polymicrobial co-infections. These findings highlight the dual utility of tNGS for diagnosing *C. neoformans* infections and identifying clinically significant polymicrobial coinfections.

Among the 22 cases with polymicrobial co-detections, the most frequent species included Epstein–Barr Virus (EBV, *n*=13), Herpes simplex virus-1 (HSV-1, *n*=8), *Staphylococcus aures* (*n*=10), *Streptococcus anginosus* (*n*=7), *Klebsiella pneumoniae* (*n*=5), *Candida albicans* (*n*=12) and *Pneumocystis jirovecii* (*n*=6) (Fig 2C). Pneumonia was diagnosed in 77.3% (17/22) of these cases. Four cases were considered to have viral pneumonia: SARS-CoV-2 infection in an 83-year-old female and an 82-year-old male; human metapneumovirus infection in a 75-year-old male; and HSV-1 pneumonia in an 87-year-old male. Four patients were diagnosed with *P. jirovecii* pneumonia (in a/an 34-year-old male, 75-year-old

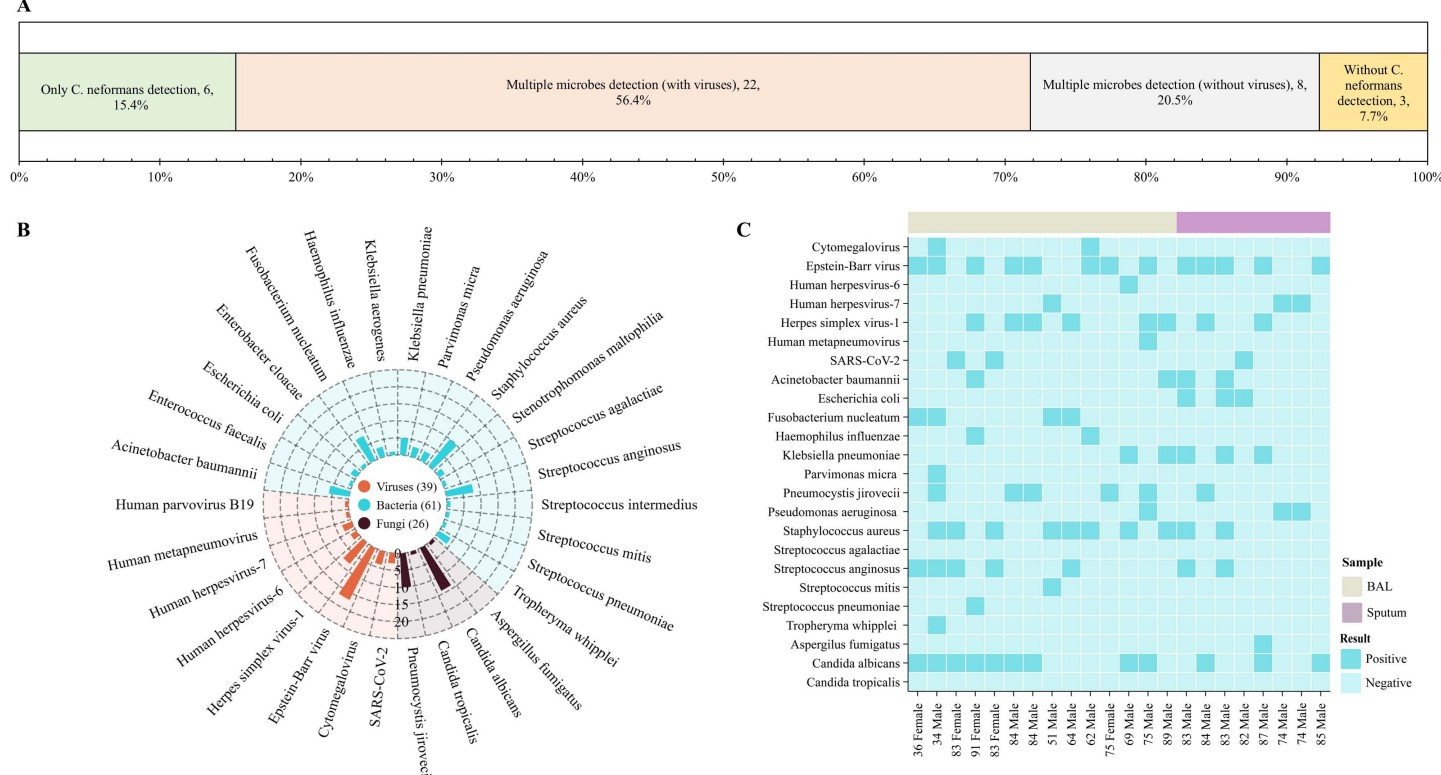

**Fig 2. Microbial Composition in Respiratory Specimens Detected by tNGS.** Panel A: Distribution of *C. neoformans* detection patterns: single pathogen detection, polymicrobial detection (with or without viruses), and specimens where *C. neoformans* was not detected; Panel B: Spectrum of microorganisms identified within the 30 polymicrobial specimens positive for *C. neoformans*. The Arabic numerals in the figure representing the number of pathogens; Panel C: Detailed co-detection profiles for the 22 patients. Patient demographics (age and sex) are shown on the lower x-axis; sample types are indicated on the upper x-axis. Pathogens are listed on the left y-axis.

female, 83-year-old female and 84-year-old male). Clinicians interpreted the tNGS detected EBV and *Candida albicans* in two elderly males (74 and 85 years) as probable sample contamination. The remaining 7 patients exhibited co-infections involving multiple pathogens.

In three cases, tNGS failed to detect *C. neoformans* in respiratory specimens. The first case was a 94-year-old female with BAL culture confirmed *C. neoformans*, polymicrobial species were detected by tNGS (including *Pseudomonas aeruginosa*, *Enterobacter cloacae*, *Acinetobacter baumannii*, *Stenotrophomonas maltophilia*, *Candida albicans*, HSV-1, human parvovirus B19, cytomegalovirus, and EBV). The other two patients, who were evaluated for cryptococcosis due to presenting with pneumonia, had negative respiratory cultures and tNGS results. The diagnosis was instead confirmed by positive serum CrAg serology. (Respiratory specimen testing results for enrolled patients are detailed in S2 Table).

## Diagnostic implication of tNGS for cryptococcosis

Among the 36 respiratory specimens testing positive for *C. neoformans* by tNGS, the median turnaround time (TAT, from specimen collection to report submission) was 23 hours (IQR: 21–43; Fig 3A). To compare TAT between methods, specimens with culture-confirmed *C. neoformans* were designated as the culture group. The tNGS group exhibited a significantly shorter median TAT than the culture group (112 hours [IQR: 77.5–120.5]; $P < 0.0001$, Mann-Whitney U test; Fig 3B).

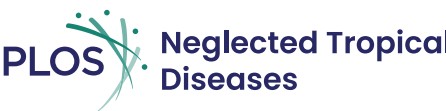

**Fig 3. Clinical implication of tNGS for pulmonary cryptococcosis.** Panel A: The distribution of TAT for the 36 tNGS tests, with the median being 23 (IQR: 21–43) hours (indicated by the light blue circles in the figure); Panel B: Dot plot comparing the difference in TAT of *C. neoformans* identification between tNGS and fungal culture; Panel C: Comparison of *C. neoformans* sequence reads detected by tNGS between patients with culture-positive versus culture-negative respiratory specimens; Panel D: Comparison of *C. neoformans* sequence reads detected by tNGS between patients with serum CrAg-positive versus CrAg-negative status. (TAT for tNGS was defined as the time span from sample collection to result reporting; TAT, turnaround time; tNGS, targeted next-generation sequencing).

Further analysis of tNGS-positive specimens revealed that the culture-confirmed subgroup had significantly higher *C. neoformans* sequence reads compared to the culture negative subgroup (median: 29,334 [IQR: 17,713–65,258] vs. 873 [IQR: 229.3–5,749]; *P* = 0.0036; Mann-Whitney U test; Fig 3C). Notably, among tNGS-positive specimens, *C. neoformans* read counts showed no significant association with serum CrAg status (CrAg-positive median: 6,317 [IQR: 633–54,129] vs. CrAg-negative median: 970.5 [IQR: 310.3–13,976]; *P* = 0.2048; Mann-Whitney U test; Fig 3D).

## Diagnostic concordance and discrepancies in cryptococcosis

Serum CrAg testing was performed for 87.2% (34/39) of patients. Among those tested, 64.7% (22/34) were positive and 35.3% (12/34) were negative. The remaining 12.8% (5/39) patients did not undergo testing.

Conventional fungal culture of BAL identified *C. neoformans* in 23.1% (9/39) of patients. Notably, 33.3% (3/9) of these culture-confirmed cases tested CrAg negative. Additional respiratory cultures identified *Candida* species in 23.1% (9/39) of patients (1 sputum and 8 BAL), while 53.8% (21/39) had culture-negative respiratory specimens (15 BAL and 6 sputum). tNGS identified *C. neoformans* in 92.3% (36/39; 95%CI: 79.1% - 98.2%) of specimens, whereas conventional fungal culture was positive in only 23.1% (9/39; 95%CI: 11.1% - 39.3%) of cases. This marked difference in sensitivity was statistically significant ($P<0.001$; two-proportion z-test, Table 2).

## Discussion

In this retrospective study, cryptococcosis cases were identified at our hospital using international diagnostic guidelines [15,20,21]. The diagnostic framework relied primarily on two pillars: (1) serum CrAg positivity, and (2) detection of *C. neoformans* in respiratory specimens via tNGS or fungal culture. While serum CrAg testing exhibits high sensitivity in serum and CSF, its specificity may be limited by cross-reactivity with organisms such as *Trichosporon asahii*, *Stomatococcus* spp. and *Capnocytophaga* spp. [5,22,23]. It is worth noting that 12 patients sought medical attention due to pneumonia (10 cases) and pulmonary nodules (2 cases). Although *C. neoformans* was identified in their respiratory specimens, their serum CrAg was negative. Furthermore, 5 patients lacking serum CrAg testing yielded negative conventional fungal cultures on respiratory samples, introducing a potential risk for false-negative diagnoses. These findings underscore the limitations of relying solely on serum CrAg and highlight the necessity for multimodal diagnostic confirmation. Among the patients with negative serum CrAg, only 25% of them had *C. neoformans* identified in their BAL by conventional fungal culture. In contrast, tNGS detected *C. neoformans* in the remaining 75%, significantly enhancing diagnostic yield. With rare exceptions (e.g., iatrogenic or zoonotic transmission), naturally acquired cryptococcosis typically originates from inhaling environmental fungal cells [24,25]. Given this well-defined transmission route, respiratory testing remains critical when serum CrAg is inconclusive, particularly in cases with clinical or radiological suspicion.

tNGS demonstrates significantly improved detection of *C. neoformans* in respiratory specimens from cryptococcosis patients compared to fungal culture. In this study, fungal culture identified *C. neoformans* in only 23.1% of respiratory specimens collected from enrolled patients. In contrast, tNGS detected *C. neoformans* in 92.3% of tested specimens, revealing a statistically significant difference in detection rates ($P<0.001$; two-proportion z-test). This high tNGS positivity rate notably contrasts with prior reports [7,10,26]. A retrospective study of BAL specimens in cryptococcosis patients reported *Cryptococcus* detection rates of 43.7% (31/71) by metagenomic-NGS (mNGS) and 28.2% (20/71) by culture [27]. While our fungal culture positivity aligns with this earlier finding, our tNGS detection rate was markedly higher. This discrepancy may be attributed to variations in populations and specimen types. Our study consisted solely of diagnosed cryptococcosis patients and utilized predominantly BAL samples ($n=30$), with fewer sputum samples ($n=6$), potentially increasing the sensitivity of tNGS. Crucially, tNGS proved valuable in culture-negative cases. Among the 36 cases where *C. neoformans* was detected by tNGS, only 25% patients had concurrent positive fungal cultures from respiratory specimens. The ability of tNGS to detect cryptococcosis in culture-negative cases represents a critical diagnostic advantage. To date, tNGS applications in cryptococcosis diagnosis remain limited to isolated case reports, with no large-scale studies

**Table 2. Diagnostic performance of tNGS versus fungal culture in respiratory specimens from Cryptococcosis patients.**

| Detection Method | True positive | Sensitivity (95% CI) | 95% Confidence Interval | Statistical Comparison |
|---|---|---|---|---|
| tNGS | 36/39 | 92.3% | 79.1%-98.2% | Reference |
| Fungal culture | 9/39 | 23.1% | 11.1%-39.3% | $P<0.001$ |

validating its clinical value [28]. While our findings position tNGS as a promising tool for early diagnosis, broader multicenter studies with larger sample sizes are essential to confirm its diagnostic efficacy and optimize its integration into clinical workflows.

Although *C. neoformans* typically appears as a round or oval yeast (4 - 6 μm diameter) with a prominent capsule (up to 30 μm thick) visible by light microscopy and is cultivable on standard media, its detection by culture often requires prolonged incubation. While growth may occur within 1–3 days under optimal conditions, identifiable colony formation can require up to one week. Consequently, cultures should be maintained for a minimum of one week when cryptococcosis is clinically suspected [29,30]. Turnaround time analysis showed that tNGS identified *C. neoformans* significantly faster than culture (median 23 h vs. 112 h; $P < 0.0001$, Mann-Whitney U test). This finding confirms that tNGS not only enhances detection rate but also drastically reduces time to diagnosis, facilitating earlier clinical intervention. This phenomenon aligns with the findings of some studies [10,31,32]. Furthermore, tNGS demonstrated superior sensitivity to culture in respiratory samples. Among 36 patients with tNGS-confirmed *C. neoformans*, concurrent fungal cultures yielded negative results for fungi in 52.8% (19/36). Notably, in 25% (9/36) of specimens where yeast growth occurred, conventional culture predominantly identified isolates as *Candida* species rather than *C. neoformans*. This evidence strongly supports tNGS as a more reliable detection modality for this pathogen. This evidence demonstrates that tNGS is a method capable of increasing the identification rate of *C. neoformans*. Supporting this observation, cases with culture-confirmed *C. neoformans* exhibited significantly higher tNGS sequencing reads compared to culture-negative cases, suggesting microbial load influences culture recovery rates. Therefore, when tNGS detects *C. neoformans*, clinicians should promptly notify microbiology laboratories to extend fungal culture incubation periods, thereby enhancing *C. neoformans* recovery. The integration of rapid tNGS screening with optimized, extended culture protocols significantly improves diagnostic accuracy and enables timely clinical management of respiratory cryptococcosis. However, despite tNGS's high sensitivity, three patients with pulmonary cryptococcosis tested negative for *C. neoformans* by tNGS (patients 12, 14, and 19). The organism was cultured from the BAL of patient 14, whereas tNGS identified only mixed flora. The remaining two patients were negative on all respiratory tests and diagnosed exclusively by serum CrAg. The most plausible explanation for these tNGS false negatives is not assay sensitivity but rather pre-analytical variables, including sample handling, that compromise the integrity or representativeness of the nucleic acid template. These results reveal that respiratory testing can fail to diagnose some cases of cryptococcosis, reinforcing the essential role of serum CrAg testing in screening for this disease.

Although CrAg testing demonstrates high sensitivity and positive rate, a small proportion of cryptococcosis patients with negative CrAg results may still be missed clinically [33–35]. In the present cohort, 30.8% (12/39) of patients lacked detectable cryptococcal antigenemia. Notably, tNGS identified *C. neoformans* in respiratory specimens from all 12 patients, enabling their definitive diagnosis. This suggests that tNGS can effectively detects cryptococcosis in the absence of serum CrAg positivity, thereby expanding screening and diagnostic capabilities, particularly among immunocompetent populations. Importantly, no significant difference in tNGS sequence reads was observed between CrAg-positive and CrAg-negative patients. This suggests that cryptococcal antigenemia may not directly correlate with fungal burden in respiratory specimens. Collectively, these findings empirically support integrating tNGS into practical diagnostic algorithms (Fig 4) and establish a foundation for future large-scale studies to optimize diagnostic protocols, enabling earlier and more precise management of cryptococcosis.

The clinical adoption of tNGS is also influenced by its economic feasibility. In our setting, the per-sample cost of tNGS (750 RMB) is approximately 7–9 times higher than that of fungal culture (85 RMB) or CrAg testing (100 RMB). However, this higher direct cost must be balanced against its superior diagnostic yield and rapid turnaround time. Batch processing can substantially reduce the per-test cost. More importantly, the comprehensive nature of tNGS can circumvent the need for a costly and protracted series of sequential tests. Although a formal cost-effectiveness analysis is required, we hypothesize that the faster time-to-diagnosis facilitates earlier targeted therapy, which may shorten hospital stays and reduce overall healthcare expenditures, potentially offsetting the higher initial laboratory cost.

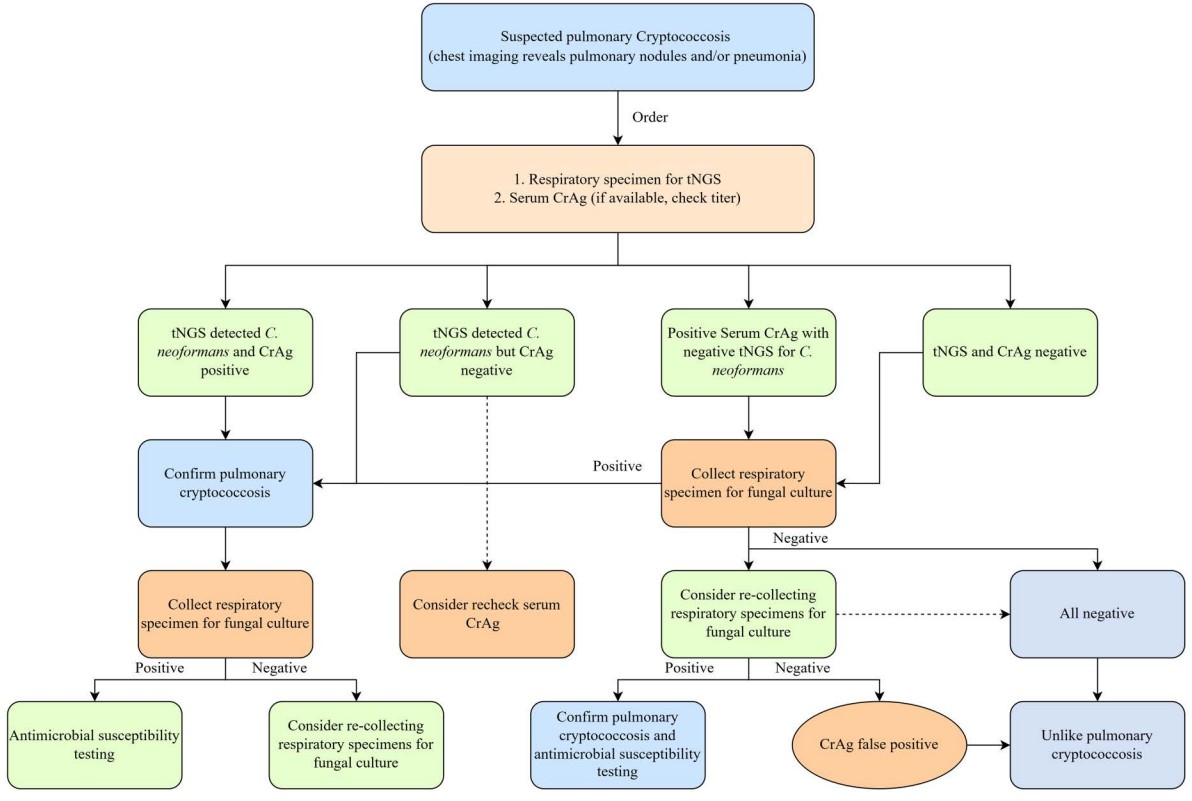

**Fig 4. A practical diagnostic algorithm for pulmonary cryptococcosis.** (tNGS: Targeted Next-Generation Sequencing, CrAg: cryptococcal antigen).

Our research has several limitations that merit consideration. First, the cohort size was small, as tNGS testing of respiratory specimens was not on all patients with suspected pulmonary cryptococcosis lesions, and serum CrAg testing was also selective. Consequently, patient low clinical suspicion may have been missed. Future studies with larger sample sizes are warranted to further evaluate the diagnostic role of tNGS in cryptococcosis. Second, the number of sequences detected by tNGS is influenced by the quality and volume of respiratory specimens. The quality and volume of specimens were not systematically evaluated here, which could introduce a potential bias in the assessment of sequence reads in tNGS. Third, because CrAg titer analysis was not routinely performed on positive patients, we cannot exclude potential false positives or assess the correlation between CrAg titers and *Cryptococcus* sequence reads derived by tNGS. Fourth, we did not analyze potential correlations between *C. neoformans* and other microorganisms detected by tNGS. Co-infecting pathogens could influence patient outcomes, warranting future investigation. Finally, routine antifungal suscep-tibility testing was not performed on cultured *C. neoformans* isolates, nor did we use tNGS to detect resistance genes or predict resistance phenotypes. Future work should explore the relationship between *Cryptococcus* resistance genes identified by tNGS and phenotypic AST results to guide more precise treatment and advance research into *C. neoformans* drug resistance.

## Conclusion

This study demonstrates that tNGS provides high sensitivity and markedly shorter turnaround time compared with conven-tional culture. These findings underscore its growing importance as a key tool in the clinical diagnosis and management of infectious diseases. As a cross-sectional, single-center study, it provides a foundation for future multicenter trials aimed

at optimizing the integration of tNGS into diagnostic workflows. Additionally, future research should explore tNGS guided resistance profiling to refine therapeutic strategies for cryptococcosis.

## Supporting information

**S1 Table. Serial of Cryptococcosis patients.**
(XLSX)

**S2 Table. The results of tNGS testing in this study.**
(XLSX)

**S3 Table. List of pathogens detected using tNGS.**
(XLSX)

**S1 Data. Supplementary Information.** Section A. Pathogens that can be detected by targeted next-generation sequencing (tNGS) in this study. Table A. Clinical and laboratory findings of enrolled patients in this study.
(DOCX)

## Acknowledgments

We thank the staff at the Department of Infections and Microbiology, The University of Hong Kong - Shenzhen Hospital for their technical support and assistance. We would also like to extend our gratitude to KingMed Diagnostics (Guangzhou) for performing tNGS detection and providing methodological support.

## Author contributions

**Conceptualization:** Chaowen Deng, Fanfan Xing.

**Data curation:** Chaowen Deng, Miaoling Qiu, Qingyan Yang, Lina Li, Fanfan Xing.

**Funding acquisition:** Chaowen Deng, Fanfan Xing.

**Investigation:** Chaowen Deng, Miaoling Qiu, Qingyan Yang, Lina Li, Fanfan Xing.

**Methodology:** Baoling Liu, Jieling Liu, Ricky Wing-Tong Lau.

**Project administration:** Chaowen Deng.

**Supervision:** Chaowen Deng, Fanfan Xing.

**Writing – original draft:** Chaowen Deng, Fanfan Xing.

**Writing – review & editing:** Chaowen Deng, Miaoling Qiu, Qingyan Yang, Lina Li, Ricky Wing-Tong Lau, Fanfan Xing.

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
