## [Decision Letter · Decision Letter 0]

22 Oct 2025

PNTD-D-25-01306

Rapid Respiratory Cryptococcosis Detection Using Targeted Next-Generation Sequencing 

Dear Dr. Deng,

Thank you for submitting your manuscript to PLOS Neglected Tropical Diseases. After careful consideration, we feel that it has merit but does not fully meet PLOS Neglected Tropical Diseases's publication criteria as it currently stands. Therefore, we invite you to submit a revised version of the manuscript that addresses the points raised during the review process.

Please submit your revised manuscript within 30 days Dec 21 2025 11:59PM. If you will need more time than this to complete your revisions, please reply to this message or contact the journal office at plosntds@plos.org. Please include the following items when submitting your revised manuscript: 

We look forward to receiving your revised manuscript.

Kind regards,

Emmanuel Siddig

Academic Editor

Marcio Rodrigues 

Section Editor

Shaden Kamhawi

co-Editor-in-Chief

Paul Brindley

co-Editor-in-Chief

 **Journal Requirements:**

At this stage, the following Authors/Authors require contributions: Chaowen Deng, Miaoling Qiu, Qingyan Yang, Lina Li, Baoling Liu, Jieling Liu, Ricky Wing-Tong Lau, and Fanfan Xing. Please ensure that the full contributions of each author are acknowledged in the 'Add/Edit/Remove Authors' section of our submission form.

2) We have noticed that you have uploaded Supporting Information files, but you have not included a list of legends. Please add a full list of legends for your Supporting Information files after the references list.

3) In the online submission form, you indicated that The datasets generated during and/or analyzed during the current study are available from the corresponding author (CD, dengchaowen08@163.com) on reasonable request.. All PLOS journals now require all data underlying the findings described in their manuscript to be freely available to other researchers, either

1. In a public repository

2. Within the manuscript itself

3. Uploaded as supplementary information.

4) Please provide a detailed Financial Disclosure statement. This is published with the article. It must therefore be completed in full sentences and contain the exact wording you wish to be published.

1) Please clarify all sources of financial support for your study. List the grants, grant numbers, and organizations that funded your study, including funding received from your institution. Please note that suppliers of material support, including research materials, should be recognized in the Acknowledgements section rather than in the Financial Disclosure

2) State the initials, alongside each funding source, of each author to receive each grant. For example: 'This work was supported by the National Institutes of Health (####### to AM; ###### to CJ) and the National Science Foundation (###### to AM).'

3) State what role the funders took in the study. If the funders had no role in your study, please state: 'The funders had no role in study design, data collection and analysis, decision to publish, or preparation of the manuscript.'

4) If any authors received a salary from any of your funders, please state which authors and which funders..

5) Your current Financial Disclosure states, 'The author(s) received no specific funding for this work.'.

However, your funding information on the submission form indicates receiving fund . 

Please indicate by return email the full and correct funding information for your study and confirm the order in which funding contributions should appear. Please be sure to indicate whether the funders played any role in the study design, data collection and analysis, decision to publish, or preparation of the manuscript.

6) Kindly revise your competing statement in the online submission form to align with the journal's style guidelines: 'The authors declare that there are no competing interests.'

**Reviewers' comments:**

Reviewer's Responses to Questions

**Key Review Criteria Required for Acceptance?**

**Methods**

-Are the objectives of the study clearly articulated with a clear testable hypothesis stated?

-Is the study design appropriate to address the stated objectives?

-Is the population clearly described and appropriate for the hypothesis being tested?

-Is the sample size sufficient to ensure adequate power to address the hypothesis being tested?

-Were correct statistical analysis used to support conclusions?

-Are there concerns about ethical or regulatory requirements being met?

Reviewer #1: (No Response)

Reviewer #2: -The objectives are generally stated but the hypothesis is not explicitly articulated. The authors should clearly define a testable hypothesis reflecting the comparative evaluation of the three diagnostic methods for cryptococcosis.

-Figure 1 and study population:

Figure 1 indicates that 80 patients with confirmed cryptococcosis were enrolled, implying that the European Organization for Research and Treatment of Cancer and the Mycoses Study Group Education and Research Consortium (EORTC/MSGERC) guideline was used as the diagnostic gold standard. This should be explicitly stated in the Methods section to clarify the reference framework for case confirmation.

While the manuscript primarily focuses on tNGS, the diagnostic performance of the other two methods evaluated is also of clinical relevance. Given that the study population (n = 39) may be insufficient to independently establish the diagnostic accuracy of tNGS, I recommend including a comparative performance analysis of all three methods using the same patient cohort. Highlighting how tNGS performs relative to established methods would enhance the clinical value and originality of the work, framing it as a true comparative diagnostic study—the main novelty identifiable from the current manuscript.

-The manuscript needs clarification regarding the study design. Although described as retrospective, lines 108–119 read as if the study were prospective. Please clearly state how samples were obtained and indicate whether they originated from a previous study, including its title or ethics reference if applicable.

-The study population is generally appropriate.

-There are inconsistencies regarding informed consent (lines 103–105 versus 186–187). Please clarify whether patient consent was obtained and confirm compliance with ethical standards.

•Line 146: Clarify the use of 0.1 M DTT for mucolysis, as 0.1% (w/v) is the more typical concentration. Provide a citation or indicate if this followed a manufacturer’s protocol.

•Line 144: Replace “commercial company laboratory” with “private laboratory” or “the laboratory of a private company.”

**Results**

-Does the analysis presented match the analysis plan?

-Are the results clearly and completely presented?

-Are the figures (Tables, Images) of sufficient quality for clarity?

Reviewer #1: (No Response)

Reviewer #2: -Lines 277–280: The two sentences appear contradictory. Clarify what was actually performed.

-The authors should report the calculated sensitivity and specificity of tNGS compared to the other diagnostic methods (or at least the gold standard) to substantiate conclusions made later in the manuscript (line 432).

-Table 1: Age is expressed as years (IQR) while other variables are n (%). Find a way to indicate this.

-Figure 4: The flowchart is confusing. Usually, boxes containing questions and others with “yes”/“no” or “positive”/”negative” should be used to denote what decision to take after an outcome. For instance, the current chart does not indicate what happens to “respiratory specimen for culture”.

-Line 289: The title “Clinical implication of tNGS for cryptococcosis” may be better phrased as “Diagnostic implication of tNGS for cryptococcosis.”

**Conclusions**

-Are the conclusions supported by the data presented?

-Are the limitations of analysis clearly described?

-Do the authors discuss how these data can be helpful to advance our understanding of the topic under study?

-Is public health relevance addressed?

Reviewer #1: (No Response)

Reviewer #2: The conclusions that tNGS has high sensitivity and specificity should fully supported by quantitative evidence in the results by specifically calculating these.

**Editorial and Data Presentation Modifications?**

Reviewer #1: (No Response)

Reviewer #2: •Title: Does not fully reflect the manuscript’s content. Recommend revising to highlight the comparative nature of the study (e.g., “Comparative Evaluation of Three Diagnostic Methods for Cryptococcosis in a Clinical Setting”).

•Line 66 should specify which organs are most commonly affected by cryptococcosis to provide clinical context.

•Line 91: Define “t” in tNGS at first mention.

•Line 92: Italicize all species names.

•Line 237: Replace aures with aureus. Ensure subsequent mentions are abbreviated properly and italicized.

•Lines 262–263: The sentence beginning “The other two cases…” is grammatically incorrect and confusing. Revise to clearly describe the findings.

•Line 349: Change to “potentially increasing the sensitivity of tNGS” instead of “potentially increased tNGS sensitivity.”

•Lines 385–392 provide a weak rationale for the observed limitation. In active infections, tNGS should detect genetic material if sample collection and processing are appropriate. The limitation is more likely due to sample handling, as discussed in lines 416–418. Revise the earlier explanation accordingly.

•Line 394: Delete “in.”

•Discussion: Reduce redundancy by removing repeated descriptions of the results. Focus instead on interpretation and implications.

•Ensure all abbreviations are defined at first use and standardized throughout the manuscript.

•Grammar and syntax should be checked carefully throughout for consistency and conciseness.

**Summary and General Comments**

Reviewer #1: (No Response)

Reviewer #2: The manuscript is clearly written with only minor errors. The objectives are generally stated but the hypothesis is not explicitly articulated. The study population is generally appropriate and the manuscript has clearly defined sample characteristics and the limitation acknowledged. The above corrections should be considered to improve the manuscript.

PLOS authors have the option to publish the peer review history of their article (what does this mean?). If published, this will include your full peer review and any attached files.

Reviewer #1: **Yes:** MAQSUD HOSSAIN

Reviewer #2: No

**Figure resubmission:**

After uploading your figures to PLOS’s NAAS tool - https://ngplosjournals.pagemajik.ai/artanalysis, NAAS will process the files provided and display the results in the 'Uploaded Files' section of the page as the processing is complete. If the uploaded figures meet our requirements (or NAAS is able to fix the files to meet our requirements), the figure will be marked as 'fixed' above. If NAAS is unable to fix the files, a red 'failed' label will appear above. When NAAS has confirmed that the figure files meet our requirements, please download the file via the download option, and include these NAAS processed figure files when submitting your revised manuscript.
---

## [Editor Report · Decision Letter 1]

12 Nov 2025

Dear Dr. Deng,

We are pleased to inform you that your manuscript 'Rapid Respiratory Cryptococcosis Detection Using Targeted Next-Generation Sequencing' has been provisionally accepted for publication in PLOS Neglected Tropical Diseases.

Best regards,

Emmanuel Siddig

Academic Editor

Marcio Rodrigues

Section Editor

Shaden Kamhawi

co-Editor-in-Chief

Paul Brindley

co-Editor-in-Chief

---

## [Editor Report · Acceptance letter]

Dear Dr. Deng,

We are delighted to inform you that your manuscript, "Rapid Respiratory Cryptococcosis Detection Using Targeted Next-Generation Sequencing," has been formally accepted for publication in PLOS Neglected Tropical Diseases.

Best regards,

Shaden Kamhawi

co-Editor-in-Chief

Paul Brindley

co-Editor-in-Chief
